# How Does Trauma Make You Sick? The Role of Attachment in Explaining Somatic Symptoms of Survivors of Childhood Trauma

**DOI:** 10.3390/healthcare12020203

**Published:** 2024-01-15

**Authors:** Paul Samuel Greenman, Alessia Renzi, Silvia Monaco, Francesca Luciani, Michela Di Trani

**Affiliations:** 1Département de Psychoéducation et de Psychologie, Université du Québec en Outaouais, Gatineau, QC J8X 3X7, Canada; paul.greenman@uqo.ca; 2Institut du Savoir Montfort, Ottawa, ON K1K 0T2, Canada; 3Department of Dynamic and Clinical Psychology and Health Studies, University of Rome, 00185 Rome, Italy; silvia.monaco@uniroma1.it (S.M.); luciani.1799082@studenti.uniroma1.it (F.L.); michela.ditrani@uniroma1.it (M.D.T.)

**Keywords:** childhood trauma, attachment, somatic symptomatology, review

## Abstract

Exposure to traumatic events during childhood is common, and the consequences for physical and mental health can be severe. Adverse childhood experiences (ACEs) such as physical abuse, sexual abuse, emotional abuse, and neglect appear to contribute to the onset and severity of a variety of somatic inflictions, including obesity, diabetes, cancer, and heart disease. The aim of this scoping review was to try to gain insight into how this might occur. Given the evidence of indirect (i.e., through unhealthy behaviours such as excessive drinking or poor eating habits) and direct (i.e., through its impact on the endocrine, immune, and cardiovascular systems as well as on the brain) effects of attachment on health, we examined the possibility that insecure attachment might contribute to the development of somatic symptoms in adult survivors of childhood trauma. Eleven studies met our inclusion criteria. Findings from this review suggest that insecure and disorganized attachment orientations are related to DNA damage, metabolic syndrome and obesity, physical pain, functional neurological disorder, and somatization in adults exposed to childhood trauma. We discuss the implications of this for the conceptualization and treatment of trauma and stress disorders.

## 1. Introduction

Traumatic experiences during childhood are varied and widespread, with consequences for mental and physical health over the lifespan. Researchers in this area have traditionally examined the long-term effects of adverse childhood experiences (ACEs) including physical abuse; emotional abuse; sexual abuse; emotional neglect; physical neglect; loss of a parent through death, divorce, or imprisonment; witnessing domestic violence; parental substance abuse; and mental illness in one or both parents [1,2]. Data from the United States of America indicate a lifetime prevalence of at least one ACE in nearly 67% of the population [1]. In a more recent investigation in the USA, Giano et al. [3] reported similar findings. Prevalence rates in European countries appear to be somewhat lower but nevertheless significant, ranging from 19% (United Kingdom) to 34% (Denmark) for one ACE [4].

It is important to note that “childhood trauma” refers to the mental consequences of an external and sudden traumatic event or a series of traumatic events. The term “complex trauma” refers to repeated and prolonged early traumatic experiences in attachment relationships [5,6,7]. Complex trauma includes sexual, physical, or emotional abuse in childhood or adolescence by a primary caregiver, or another member of the family or social group to which the child belongs. About 80% of maltreatment experiences occur within the household, perpetrated by parents or adults who play a significant role in the child’s life [8,9]. In both cases, the individual is exposed to traumatic interpersonal events at a young age, which can manifest themselves later in life as a distinct psychological syndrome. Our focus in this paper is on exposure to traumatic experiences during childhood, which may or may not culminate in the onset of diagnosable symptoms.

Common health problems in survivors of traumatic experiences during childhood are both mental and physical. In terms of mental health, they are more likely than people who are not exposed to trauma to develop symptoms of depression, anxiety, post-traumatic stress, and eating disorders [2,4,10,11]. Physical health problems include addictions (i.e., illicit drug use, heavy alcohol use, smoking) and a variety of somatic conditions such as obesity, diabetes, cancer, and cardiovascular disease, to name a few [2,4]. Although the mechanisms of the deleterious effects of childhood trauma on mental and physical health are becoming increasingly clear (e.g., changes in self-concept and notions of self-worth, alterations in the production of corticosteroids by the hypothalamic-pituitary-adrenal axis, interactions between genetic predispositions and trauma in the environment) [12], there is still a great deal to learn about exactly how exposure to traumatic events during childhood can lead to illness, and particularly to physical disease.

Attachment theory might provide some answers to this. In adulthood, “secure attachment” involves positive representations of the self and others, which lead to a tendency to actively seek out and to receive emotional support [13]. “Insecure attachment” can involve negative views of the self and positive perceptions of others, as is the case with preoccupied attachment. It can also entail positive perceptions of the self and negative views of others, as is the case with dismissing-avoidant attachment, or it can represent negative views of the self and negative views of others, as is the case with fearful-avoidant attachment [14,15]. Fearful-avoidant attachment is sometimes referred to as “disorganized attachment”, and it has been linked to experiences of trauma and maltreatment by caregivers during childhood [16]. In terms of emotion regulation, preoccupied attachment generally comprises hyperactivating emotion-regulation strategies such as clinging behaviour and constant reassurance-seeking [13]. Avoidant or dismissing attachment implies hypoactivating emotion-regulation strategies such as minimizing the significance of the problem that triggers attachment behaviour along with the importance of close emotional connections. In general, secure attachment is a predictor of positive mental and physical health outcomes, whereas insecure attachment poses a health risk (see [13] for a review).

Theorists in recent years have proposed attachment theory as a framework for understanding susceptibility to chronic physical illnesses and to poor health over the long term. There is evidence of indirect (i.e., through unhealthy behaviours such as excessive drinking or poor eating habits) and direct (i.e., through its impact on the endocrine, immune, and cardiovascular systems as well as on the brain) effects of attachment on health; see [17,18,19]. Adverse relational experiences in childhood appear to favour the development of an insecure attachment pattern, which in turn can make the individual more vulnerable to somatization [20]. There is a substantial quantity of work in the literature on the associations between two out of the three constructs (i.e., attachment-somatic symptoms, attachment and childhood trauma, childhood trauma and somatic symptoms). However, to our knowledge, there is a paucity of studies exploring the relations among childhood trauma, attachment, and somatic symptoms together.

In the light of above considerations, the primary aim of this review was to investigate whether and how attachment orientation might affect the relationship between childhood trauma and somatic symptoms in adulthood. We also sought, more generally, to discover how researchers have conceptualized the relationship between these variables. This knowledge will also be helpful in orienting treatment.

For the purposes of this scoping review, we considered “somatic symptoms” to be any pathological condition, including both physical diseases with a clear physiological etiology and “somatic symptom and related disorders” as defined in the fifth edition of the Diagnostic and Statistical Manual of Mental Disorders (DSM-5) (American Psychiatric Association, 2013, Washington, DC, USA). These are physical problems that do not have an identifiable physiological origin, causing significant stress and functional impairment to the individual. Thus, somatic symptoms include all typologies of physical health outcomes identified by the respective studies’ authors as a somatic outcome. The choice of the term “somatic symptoms” was considered to be a good option because it is both sufficiently specific and appropriately general for identifying studies on a variety of somatic conditions. Thus, the focus in this initial review was wide. The purpose of the present investigation was first to clarify the nature of the link between childhood trauma, attachment, and somatic symptoms in general, and then to provide hypothesis useful to better understand the mechanisms linking childhood trauma to the onset of somatic disturbances in adulthood.

## 2. Methods

### 2.1. Search Strategy and Selection Criteria

We elected to conduct a scoping review to answer the research question “How might childhood trauma, attachment, and somatic symptoms in adulthood be related?”. This method has proven useful in identifying the type of evidence available in a given field of research, clarifying the key concepts and definitions present in the literature, examining how research is conducted on a particular topic, identifying the main characteristics or factors related to a concept, and analyzing gaps in knowledge [21,22].

Khalil and colleagues [23] have outlined a series of steps to increase the pertinence and precision of scoping reviews, which we followed in the present article. These include (1) the formulation of a research question, (2) identifying relevant studies using an inclusive group approach, (3) presentation of data in a narrative and tabular format, and (4) tracking data and collecting results to identify implications.

### 2.2. The Search

The first phase of the search for relevant articles was carried out in the main psychological databases: PsycInfo, Scopus, and Pubmed. Initially, the following keywords were used: [childhood trauma] + [attachment] + [somatic symptoms]. However, given the relative dearth of material that emerged, we decided to expand the search terms by adding an asterisk (truncation symbol) to the keywords to find documents that contained the word in its different forms. Thus, the final keywords were: [childhood trauma*] + [attachment*] + [soma*]. There were no limitations with respect to the years of publication of the research works, nor was any age range or specific research design specified. This enabled us to collect the maximum amount of information available in the existing literature.

A principal inclusion criterion was that there be a specific measurement scale for each of the variables of interest (in the case of physical illness, of course, these were often clinical samples, which therefore did not require further investigation). In order to be included in this review, studies had to meet the following inclusion criteria: (1) the presence of attachment-related variables according to attachment theory as outlined by Bowlby [24], (2) the analysis of exposure to traumatic events during childhood, (3) the measurement of somatic symptoms, (4) the use of valid measurement scales for each variable under investigation, and (5) the inclusion of adults 18 years of age and older.

Moreover, the search was limited to English-language journal articles and was adapted for each database as necessary.

Two judges (SM and FL) independently analyzed the manuscripts and decided on the eligibility or the exclusion of a particular paper. In case of disagreement, a third independent judge (AR) evaluated the material, and a final agreement was reached. The inter-rater reliability was good (Cohen’s k = 0.76). Subsequently, the three judges extracted the relevant information from all of the eligible manuscripts via thematic analysis.

Thus, our working group identified 22 studies that met the inclusion criterion by referring to the title, abstract, and method of each article, and eliminating those that appeared more than once in the various database searches.

In a second phase of our search, we examined the reference lists of the articles that were retained, as per the suggestions of Hepplestone and colleagues [25]. This generated another 14 articles that met the inclusion criterion. After a thorough reading of the full text of the 36 articles identified, it was decided to exclude 24 of them for one or more of the following reasons: (a) There was no mention of somatic disease in the presentation of the results of the research; (b) early trauma and attachment were mentioned in the method but not presented in the results or discussed in the conclusions; (c) traumatic events experienced throughout life were included in the sample; (d) somatoform dissociation, which does not constitute somatic pathology, was the sole somatic variable included the study; and (e) the researchers did not use established psychometric tools to measure constructs of interest. In total, 11 articles were included in the review (Figure 1).

## 3. Results

In this section we provide an overview of the 11 studies selected for this scoping review, grouped according to the pathology examined by the authors (see Table 1).

### 3.1. DNA Damage

Bergholz and colleagues [26] investigated whether psychological factors such as attachment and mentalization could modulate the link between childhood trauma and DNA damage. They explain that double-strand DNA breakage (DSB) is the most serious injury to genomic integrity, that a direct quantitative indicator of this damage is the phosphorylation of histone 2AX (γH2AX), and that immunofluorescence visualization of γH2AX outbreaks has established itself as a highly sensitive marker for DSBs. Begholz and collaborators isolated peripheral lymphocytes from the blood of healthy controls and from psychiatric patients with a history of childhood trauma. The psychiatric patients were divided into a group with severe trauma and a group with mild trauma. The three groups (controls, severe trauma, and mild trauma) were compared with respect to the amount of γH2AX per cell and the percentage of peripheral lymphocytes containing γH2AX. The control group showed the least γH2AX per cell, and the lowest percentage of peripheral lymphocytes containing γH2AX, followed by patients with mild trauma and then those with severe trauma. In other words, patients with complex childhood trauma showed a significantly higher level of DSB in peripheral lymphocytes and thus significantly greater DNA damage than did participants in the other two groups.

To clarify whether psychological factors influenced the effect of childhood trauma, the authors examined links between attachment style, motional activation, and the percentage of γH2AX. They found that higher levels of emotional awareness, or “mentalization”, correlated with a lower percentage of γH2AX per cell, while avoidant attachment correlated with increased markers of DNA damage in all groups, especially in patients with complex childhood trauma [26]. In fact, in this group of patients, those with avoidant attachment showed the highest percentage of γH2AX per cell. This led the researchers to conclude that psychosocial factors, such as attachment to primary figures or the ability to symbolize emotions, can influence the level of DNA damage in people with complex childhood trauma. In general, these data underscore the importance of early emotional experiences in genetically determining health status in adulthood, providing the basis for explaining why highly traumatized individuals suffer from a higher incidence of genetically determined diseases such as cancer [27].

### 3.2. Metabolic Syndrome and Obesity

Two studies of metabolic syndrome (MetS) and obesity were included in our review [28,29]. Davis and colleagues hypothesized that adult attachment and trauma would predict health behaviour and indicators of metabolic syndrome (MetS). As expected, results indicated that incoherent narratives (a marker of attachment insecurity) and the presence of unresolved trauma were directly linked to a greater number of MetS components. In addition to this direct effect, the authors also highlight indirect pathways from childhood adversity to MetS through smoking, symptoms of depression, and impaired family functioning, the latter appearing to have led to poor eating habits.

D’Argenio and colleagues focused solely on obesity, exploring whether in addition to sexual or physical abuse, less severe forms of childhood stress (e.g., separation from a parent, marital conflicts, psychiatric illness of a parent, etc.) were also linked to the development of obesity later in life and whether the association between early trauma and adult obesity was explained by “psychological dysfunction”, including anxious attachment. They analyzed the prevalence and severity of different types of early traumatic events, assessed the presence of coexisting psychiatric disorders, and measured attachment style in a sample of 200 participants, whom they divided into three groups: non-obese, obese, and obese with a psychiatric diagnosis. Participants who reported more severe early trauma were more likely to be obese at the time of experimentation than were participants with less severe early trauma. The exclusion of participants who had suffered physical or sexual abuse did not change outcomes. This confirms that early exposure to traumatic events with varying degrees of severity is associated with a high risk of obesity in adulthood. The authors also noted a strong association between anxious attachment and obesity, although early trauma remained a significant and independent predictor of obesity even when anxious attachment was taken into account. In summary, the severity of childhood trauma predicted adult obesity above and beyond the influence of psychiatric diagnosis and anxious attachment.

### 3.3. Physical Pain

Participants in the two studies of childhood trauma and physical pain that met our inclusion criteria [30,31] had a diagnosis of somatoform pain disorder (SPD), which has more recently replaced by the term somatic symptom disorder (SSD; APA, 2013). In the first study by Nacak and colleagues (2017) [31], researchers investigated the relationship between SPD, frequency of traumatic experiences in childhood and throughout life, and attachment style. The results showed that insecure attachment was significantly more prevalent in patients with SPD (60%) than it was in healthy participants (14%). In addition to insecure attachment style, other highly predictive factors of SPD were the number of traumatic events and the presence of symptoms of depression. Patients with SPD reported a higher number of traumatic events than did healthy controls, regardless of the type of trauma; 70% of them reported three or more traumatic events in their lives, whereas healthy participants mostly reported having experienced only one (40%). In terms of childhood adversity, SPD patients scored significantly higher on all subscales of the CTQ. Finally, an additional 87% of patients with SPD reported significant comorbid depression. The findings of Nacak and colleagues demonstrate that an accrual of traumatic experiences during childhood and adolescence, along with insecure attachment, are related to the development of chronic pain with or without identifiable physiological causes. In a further study, Nacak and collaborators [30] investigated the predictive power of an original variable, Rejection Sensitivity (RS), in a comparison of individuals with and without a diagnosis of SPD. They define RS as the anticipation of rejection and an increase in discomfort in response to perceived rejection. Results indicated that patients with SPD had significantly higher RS scores than did healthy controls. In addition, regardless of the presence of SPD, individuals with insecure attachment had significantly higher RS scores than did individuals with a secure attachment orientation.

Consistent with expectations and with previous research [32,33], Nacak and colleagues (2021) [30] also found that higher levels of depressive symptoms, traumatic events, childhood adversity, and insecure attachment predicted higher levels of RS in both patients and controls. Thus, attachment insecurity appears to be related not only to mental health, but also to the interpretation of socially ambiguous situations. These results extend Nacak and collaborators’ (2017) [31] earlier findings; they suggest that patients with SPD may have difficulties in social situations, particularly in situations of perceived rejection, and they indicate the importance of considering RS in clinical practice with this group of patients.

### 3.4. Functional Neurological Disorder

Functional neurological disorder (FND), also known as conversion disorder, is a common neuropsychiatric condition characterized by neurological symptoms for which no physical explanation can be identified, including weakness, difficulty walking, non-epileptic seizures, paralysis, tremors, and sensory deficits [34].

### 3.5. Motor FND

Williams and colleagues [35] conducted an investigation of 56 patients with motor FND. Fearful attachment style was associated with adverse life events during childhood, alexithymia (i.e., difficulty identifying and expression emotion), dissociation, depression, anxiety, reduced ability to cope with stress, and the severity of functional neurological symptoms [35]. These findings affirm existing evidence of the relation between fearful attachment and early traumatic experiences in people who experience psychogenic nonepileptic seizures (PNES) and in patients with SSD [31,36]. Moreover, although not all individuals with FND report suffering adverse events, these represent a well-documented risk factor for FND and they are also associated with symptom severity [37,38,39].

### 3.6. PNES (NEAD)

Two other studies included in this review [36,40] featured a focus on PNES, which is also known as nonepileptic attack disorder (NEAD). PNES are sudden, uncontrollable changes in consciousness, movement, and perception, similar to epileptic seizures. However, they are not accompanied by the electrophysiological changes in the brain that typify epilepsy. There seems to be a consensus that psychological components play a crucial etiological role in most patients with PNES, e.g., [41,42,43]. Gerhardt and colleagues (2021) [40] explored childhood trauma and attachment in patients with PNES. Their results showed significantly fewer secure attachment classifications in the PNES patient group than in the control group, with 43% vs. 23%, PNES and controls, respectively, classified as unresolved/disorganized. They also found unresolved attachment to be significantly associated with personality disorders and childhood emotional abuse. Thus, childhood trauma and insecure attachment, particularly unresolved/disorganized attachment, appear to be key pieces of the PNES puzzle. Holman and colleagues (2008) hypothesized that patients with PNES would demonstrate a greater prevalence of insecure attachment and a higher incidence of traumatic childhood experiences than would individuals with epilepsy. A significant difference in attachment style between the two groups emerged, with a predominance of fearful attachment in the PNES group and a predominance of secure attachment in the epilepsy group [36]. Abuse and neglect were also significantly more common in patients with NEAD than they were in those with epilepsy. Thus, both early traumatic experiences and fearful attachment predicted NEAD in this investigation. Regarding psychopathology, participants in the PNES group reported significantly more anxiety than did participants in the epilepsy group. Although psychopathology is associated with a fearful attachment style [44,45], in this sample attachment was still associated with PNES even after controlling for anxiety and dysthymia, suggesting a specific link between the two [36].

In conclusion, the results seem to sustain a link between attachment, PNES, and early traumatic experiences. Indeed, in the international literature it is known that disorganized attachment is closely associated with abuse and neglect, as well as the development of dissociative psychopathology [46,47]. Thus, attachment dimensions and childhood traumatic experience may have a direct or an indirect effect on the development of dissociative pathology. NEAD is often considered to be a form of dissociative disorder and is classified as such in ICD-10.

### 3.7. Somatization

Our search generated three studies of somatization that met our inclusion criteria [48,49,50]. Waldinger and colleagues tested whether insecure attachment mediated the link between childhood trauma and somatization in adulthood. One hundred and one couples participated and findings showed that childhood trauma was associated with higher levels of somatization and insecure attachment, with different patterns for men and women [50]. For women, insecure attachment completely mediated the link between childhood trauma and somatization. For men, however, there was no such mediation; both childhood trauma and insecure attachment contributed independently to SSI scores. These results suggest that, for men, both trauma and attachment may be important independent predictors of somatization, and for women, that childhood trauma tends to exert its effects on somatization indirectly through attachment. These gender differences may be related to the types of abuse suffered by each group. For example, women were three times more likely than men to have been sexually abused. Sexual abuse has a major impact on attachment security, particularly when the perpetrators are attachment figures [51].

Brianda and colleagues investigated, in a group of young women, the possible effect of the quality of romantic attachment on the link between adverse experiences in childhood and somatic symptoms in adulthood. Their results confirm a relationship between emotional maltreatment in childhood and somatic symptoms in adulthood among women [48]. There was also a direct link between self-reported emotional abuse during childhood and high levels of anxiety and avoidance in couple relationships. Finally, romantic attachment mediated the relation between childhood emotional abuse and symptoms of somatic disorders in adulthood. This suggests that insecure romantic attachment might strengthen the link between early emotional trauma and later somatization.

Caplan and colleagues examined the link between childhood abuse and inflammatory bowel disease (IBD)-related outcomes in 205 patients (disease activity and quality of life), and the degree to which insecure attachment moderated that relationship. Regarding IBD-related quality of life, patients with less severe abuse and lower levels of avoidant attachment reported the highest levels of quality of life in the sample, whereas patients with higher levels of insecure attachment (avoidant and anxious) reported the lowest levels of quality of life, regardless of the severity of the abuse they indicated having suffered during childhood [49]. As for disease activity, patients with less severe abuse and lower levels of avoidant attachment had the lowest scores. In contrast, patients with high levels of avoidant attachment demonstrated the highest levels of UC-related disease activity, regardless of the severity of abuse. There was no significant effect of anxious attachment on the association between childhood abuse and UC-related disease activity. In addition, childhood abuse and attachment were not significantly associated with disease activity in CD patients. In summary, an avoidant attachment orientation was found to moderate the relationship between childhood abuse and quality of life in all patients (CD and UC), and between childhood abuse and disease activity only in UC patients. The overall findings of this study confirm that adult attachment can moderate the link between childhood abuse and IBD-related outcomes, influencing quality of life and disease activity [49]. However, different types of insecure attachment might have different effects on these relationships. For IBD patients, avoidant attachment style appears to be the most relevant predictor of both lower quality of life and worse disease activity.

Finally, the interaction between childhood abuse and avoidant attachment was significantly associated with disease activity only in patients with Crohn’s disease, and not in those with ulcerative colitis [49]. Although these two pathologies share many clinical symptoms, Crohn’s disease is often a more complicated and serious condition, accompanied by greater psychological distress, worse quality of life, and greater use of health care services [52].

In summary, the primary finding emerging from this scoping review is that attachment appears to be related to somatic symptoms in people who have experienced childhood trauma, such that insecure attachment orientations generally predict the presence or intensity of somatic symptoms. This is true for survivors of physical, emotional, and sexual abuse; emotional neglect; and a host of other adverse childhood events. The physical symptoms investigated include DNA damage, metabolic syndrome and obesity, functional neurological disorder, and somatization.

**Table 1 healthcare-12-00203-t001:** The 11 articles discussed in the scoping review.

Author and Year	Title	Scale *	Participants’ Information	Results
DNA Damage				
Bergholz et al., 2017 [26]	DNA damage in lymphocytes of patients suffering from complex traumatization	CTQ, AAP, LEAS	Clinical group *n* = 40 (25 F; 15 M; mean age = 41.8; sd = 12.5) Control group *n* = 20 (14 F; 6 M; mean age = 44; sd = 13.1)	Patients with severe early trauma showed greater damage to DNA than did patients with mild early trauma and healthy controls. Patients with avoidant attachment showed greater DNA damage than did patients with secure or anxious attachment.
Metabolic Syndrome and Obesity				
Davis et al., 2014 [29]	Adult attachment interview discourse patterns predict metabolic syndrome in midlife	ELS, AAI, SCID-I, SAS, BDI, FFQ	*n* = 215 (112 F; 103 M; mean age = 45.9; sd = 3.3)	Less consistency, unresolved attachment, and idealization correlated with more MetS components; indirect paths led from childhood adversity to MetS.
D’Argenio et al., 2009 [28]	Early trauma and adult obesity: Is psychological dysfunction the mediating mechanism?	ETLE, RQ, SCID	Obese group *n* = 65 (23 F; 42 M; mean age = 40.4; sd = 12) Obese + Psychiatric group *n* = 85 (22 F; 64 M; mean age = 39.1; sd = 11.5) Control group *n* = 50 (426 F; 124 M; mean age = 32.6; sd = 11.2)	Not only sexual and physical abuse, but also minor traumas in childhood were associated with obesity in adulthood; strong association between anxious attachment and obesity, but childhood trauma predicted obesity independently from attachment.
Physical Pain				
Nacak et al., 2017 [31]	Insecure attachment style and cumulative traumatic life events in patients with somatoform pain disorder: A cross-sectional study	CTQ, ETI, RSQ, PHQ-9, PHQ-15, SCID-I	Clinical group *n* = 65 (45 F; 20 M; mean age 47.5; sd = 10.6) Control group *n* = 65 (49 F; 16 M; mean age = 43.9, sd = 11.8)	Patients with SPD report greater insecure attachment than healthy controls; patients with SPD report increased frequency of traumatic events compared to healthy controls.
Nacak et al., 2021 [30]	High rejection sensitivity in patients wit hsomatoform pain disorder	CTQ, ETI, RS, PHQ-9, PHQ-15, SCID, Rejection Sensitivity Questionnaire	Clinical group *n* = 65 (45 F; 20 M; mean age = 47.5 sd = 10.6) Control group *n* = 65 (49 F; 16 M; mean age = 43.9 sd = 11.8)	Insecure attachment and related childhood adversities predicted higher levels of RS, regardless of the presence of SPD.
Functional Neurological Disorder				
Williams et al., 2019 [35]	Fearful attachment linked to childhood abuse, alexithymia, and depression in motor functional neurological disorders	CTQ, LEC-5, RSQ, SF-36, PHQ-15, PTSD-CL5, SOMS:CD, DES, TAS, BIS, SDQ, CD-RISC, STAI-T, NEO, BDI	*n* = 56 (41 F; 15 M; mean age = 40.2; sd = 13)	In patients with motor FND, fearful attachment correlated with adverse events and greater severity of symptoms.
Gerhardt et al., 2021 [40]	Insecure and unresolved/disorganized attachment in patients with psychogenic nonepileptic seizures	CTQ, AAP, PHQ-9, SDQ, SCID-II-WHY	Clinical group *n* = 44 (34 F; 10 M; mean age = 37.3; sd = 12) Control group *n* = 44 (34 F; 10 M; mean age = 37.3; sd = 12.3)	Patients with PNES reported more insecure attachment and unresolved resolved attachment than did healthy controls; in the patients with PNES, unresolved attachment was associated with childhood emotional abuse.
Holman et al., 2008 [36]	Adult attachment style and childhood interpersonal trauma in non-epileptic attack disorder	FBQ, RSQ, BDI, BAI, MCMI-III	NEAD group *n* = 17 (14 F; 3 M; mean age = 36.2; sd not available) Epilepsy group *n* = 26 (20 F; 6 M; mean age = 38.4; sd not available)	Patients with NEAD reported more insecure attachment and fearful attachment than did patients with epilepsy; patients with NEAD reported more traumatic experiences than did patients with epilepsy.
Somatization				
Waldinger et al., 2006 [50]	Mapping the road from childhood trauma to adult somatization: the role of attachment	CTQ, RSQ, SSI, BDI, CTS	*n* = 101 couples (101 F; 101 M; women mean age = 31.6; sd = 8.6; men mean age = 33.2; sd = 8.8)	In women, insecure attachment mediated the relationship between childhood trauma and somatization. In men, insecure attachment and childhood trauma predicted somatization independently.
Brianda et al., 2017 [48]	Emotional abuse and somatic symptoms in young adulthood. The mediating role of the romantic attachment style in a population female	CTQ-SF, ECR-R, SCL-90-R	*n* = 346 F (mean age = 23.2; sd not available)	Childhood emotional abuse correlated with somatic symptoms. Childhood emotional maltreatment correlated with avoidant attachment. Dysfunctional attachment strengthened the relationship between childhood maltreatment and somatic symptoms.
Caplan et al., 2014 [49]	Attachment, childhood abuse, and IBD-related quality of life and disease activity outcomes	CMH, ECR-R, HBI, IBDQ, MSS, UCLA	*n* = 193 (88 F; 105 M; mean age = 46.3; sd = 14.2)	Avoidant attachment moderated the relationship between childhood abuse and QOL in IBD patients, and between childhood abuse and disease activity in UC patients.

* Note: F (female); M (male); AAP (Adult Attachment Projective Picture System); BAI (Beck Anxiety Inventory); BDI (Beck Depression Inventory); BIS (Barrett Impulsivity Scale); CD-RISC (Connor–Davidson Resilience Scale); CMH (Child Maltreatment History Self-Report); CTQ (Childhood Trauma Questionnaire); CTQ-SF (Childhood Trauma Questionnaire–Short Form); CTS (Conflict Tactics Scale); DES (Dissociative Experiences Scale); ECR-R (Experiences in Close Relationships—Revised); ELS (Evaluation of Lifetime Stressors interview); ETI (Essen Trauma Inventory); ETLE (Early Traumatic Life Events); FBQ (Family Background Questionnaire); FFQ (Block Food Frequency Questionnaire); HBI (Harvey–Bradshaw Index); IBDQ (Inflammatory Bowel Disease Questionnaire); LEAS (Levels of Emotional Awareness Scale); LEC-5 (Life Events Checklist-5); MCMI-III (Millon Clinical Multiaxial Inventory); MSS (Mayo Scoring System); NEO (NEO Five-Factor Inventory-3); PHQ-9 (Patient Health Questionnaire 9); PHQ-15 (Patient Health Questionnaire 15); PTSD-CL5 (Posttraumatic Stress Disorder Checklist-5); RQ (Relationship Questionnaire); RSQ (Relationship Scales Questionnaire); SAS (Social Adjustment Scale); SCID-I (Structured Clinical Interview DSM IV-R Non-Patient Version Axis 1); SCID-II-WHY (Structured Clinical Interview for DSM-IV Axis II Disorders, Patient Questionnaire); SCL-90-R (Symptom Check List Revised); SDQ (Somatoform Dissociation Questionnaire); SF-36 (Short Form Health Survey); SOMS: CD (Screening for Somatoform Symptoms Conversion Disorder Subscale); SSI (Somatic Symptom Inventory); STAI-T (Spielberger State-Trait Anxiety Inventory); TAS (Toronto Alexithymia Scale); UCLA (UCLA Social Support Inventory).

## 4. Discussion

The findings of this review are commensurate with current theory and with the results of a steadily increasing number of studies that demonstrate the health benefits of secure attachment and the potential dangers of insecure attachment. According to the Social Baseline Theory (SBT), the human brain has evolved in such a way as to function optimally only when social resources that assist in the regulation of emotional and physiological processes are present. This “co-regulation” is thought to reduce allostatic load and to permit the allocation of cognitive and biological resources to tasks necessary for survival (e.g., memory, work, problem solving, avoidance of danger). From this perspective, trauma experienced during childhood might compromise a person’s perceived access to social and emotional support, which would heighten their subjective sense of emotional distress and lead to a dysregulated emotional state. The stress and strain on the nervous and immune systems engendered by such dysregulation could have direct effects on health, and they might also lead to behaviours such as excessive alcohol consumption, overeating, or cigarette smoking that also put one’s health at risk.

According to the broader international scientific literature, the theorized association among childhood trauma, attachment, and somatic symptoms in adulthood is supported by the findings of several previous studies showing: (1) a moderating effect of an insecure-dismissing attachment orientation on the relation between the number of adverse childhood experiences and cellular aging [53]; (2) a low cardiometabolic risk (e.g., high blood pressure) at midlife for people who perceive their childhood caregivers to be emotionally accessible [54]; (3) increases in inflammation (i.e., levels of C-reactive protein) in children who demonstrate insecure attachment to caregivers [55]; (4) a mediating effect of attachment insecurity on the relationship between maltreatment by parents during childhood and health problems in adolescence [56]; (5) a link between attachment anxiety and avoidance in adulthood and the presence of the inflammatory marker interleukin-6 [57]; and (6) the prediction of anxiety, depression, blood glucose, and glycated hemoglobin in patients participating in cardiac rehabilitation [58]. There is also evidence that insecure attachment (preoccupied, dismissing, or fearful) predicts loneliness [59], the harmful health effects of which have been well documented for over a decade [60,61].

The results of this scoping review extend previous findings by confirming an association between attachment, childhood trauma, and somatic symptoms in adulthood. Al-though it is not possible to draw conclusions on the direction of these associations because of the cross-sectional typology of the studies explored, it is possible to generate some hypotheses. Indeed, it is plausible to expect that attachment patterns might play a mediating role in the association between childhood traumatic experiences and somatic symptoms in adulthood. In this light, it can be noted that if the attachment relationship assures a secure context, despite the presence of early traumatic events, the long-term effects of trauma on physical health would likely be mitigated. On the other hand, if adverse childhood experiences involve an insecure attachment, this could represent a risk factor for the negative consequences of an early traumatic experience on health problems both in childhood and adulthood. This hypothesized mediating role of attachment has theoretical support in different theories such as Social Baseline Theory (SBT) or Wilma Bucci’s Multiple Code Theory [62,63,64] that supports how traumatic experiences may provoke a disconnection within different level of information elaboration with negative consequences on health [65].

## 5. Clinical Implications and Conclusions

We have suggested that insecure attachment might contribute to the ineffective regulation of the distress and negative affect engendered by traumatic experiences during childhood and thus put people at risk for future health problems. Unfortunately, childhood trauma is a major risk factor for insecure attachment, e.g., [51,66,67]. It therefore seems important for clinical interventions to target attachment relationships directly, with the aim of helping people establish secure bonds to significant others in order to restore emotional balance and physiological equilibrium. Emotionally focused therapy (EFT) for individuals, couples, and families is an approach that focuses on fostering and strengthening emotional connections in couple therapy, family therapy, individual therapy, and group interventions [68] and as such might be particularly suited to addressing the link between trauma, attachment, and health [18]. In fact, EFT-based interventions have already been applied in the treatment of individuals with cancer [69] and heart disease [70], with a randomized, controlled trial of the relationship education program Healing Hearts Together currently underway.

Attachment appears to be associated with the health outcomes of people who endure traumatic childhood experiences. Cross-sectional designs constitute a respectable first step, but they must be followed by longitudinal studies in order to establish causality and to clarify the potential influence of variables other than attachment on the link between childhood trauma and somatic symptoms. This review does not address specific diseases (e.g., cancer, diabetes, coronary artery disease); it will be important to discover whether the same pattern emerges in investigations of trauma, attachment, and well-defined clinical syndromes. Given the potential for attachment to have an effect on the long-term mental and physical health of people who endure childhood trauma, it would make sense for clinical interventions to focus on the strengthening of attachment security.

## 6. Limitations and Future Directions

It is important to note the limitations of the studies reviewed here. First, they are all cross-sectional investigations that may show associations among variables, but they do not allow for conclusions about causal relationships between the different constructs.

The choice of including in the present narrative review only English-language studies collected from established databases may be considered a further limit since potentially relevant studies written in different languages or contained in the grey literature have been not evaluated. Another limitation is the preponderance of self-report measures of attachment and trauma. The self-report measures present several limitations (such as social desirability bias, introspective capabilities bias, over/under-evaluation of the dimensions, etc.), especially for the retrospective investigation of a past phenomenon (i.e., traumatic experiences in childhood). Moreover, the studies analyzed use different methods to evaluate attachment and childhood traumatic events (e.g., questionnaires vs. interview, or different questionnaires) consistently with their theoretical frameworks. Although this aspect represents a limitation for the present review, the studies’ results appear mostly consistent, confirming the hypothesis of a relationship between the analyzed constructs.

In addition, the sample sizes of the cited works are insufficient to generalize findings, especially for clinical groups which appear rather heterogeneous with respect to the somatic disorders reported. Moreover, no studies analyzed the relations among trauma, attachment, and pathology during early adulthood, since the samples included in the studies presents a mean age of about 40 years, and no differences related to gender were explored. Further studies should address the question of the role of attachment in the link between trauma and health outcomes with longitudinal designs that demonstrate changes over time in the variables of interest and pay more attention to gender differences. It should include all three constructs (trauma, attachment, and health) with sources other than self-report questionnaires, such as clinician-report measures like the Adult Attachment Interview (AAI) [71], the Cameron Complex Trauma Interview (CCTI) [72], in which participants’ responses are evaluated by trained clinicians according to validated guidelines, and biophysical markers of health (e.g., cortisol, heart-rate variability, glycemic control, etc.). Moreover, it might be useful in future research to focus on specific pathologies (e.g., cardiovascular disease, diabetes, cancer) and to use them as search terms to enrich the exploration in this field.

The present scoping review offers an overview of associations between early traumatic experiences, attachment, and health consequences, allowing us to hypothesize attachment as a potential mechanism of the effects that childhood trauma can have on somatic symptoms later in life. Thus, the present findings seem to support the possibility of attachment playing a role in the health manifestations of childhood trauma. The next step will be to apply and test this model with specific symptoms and diseases (e.g., diabetes, heart disease, cancer).

## Figures and Tables

**Figure 1 healthcare-12-00203-f001:**
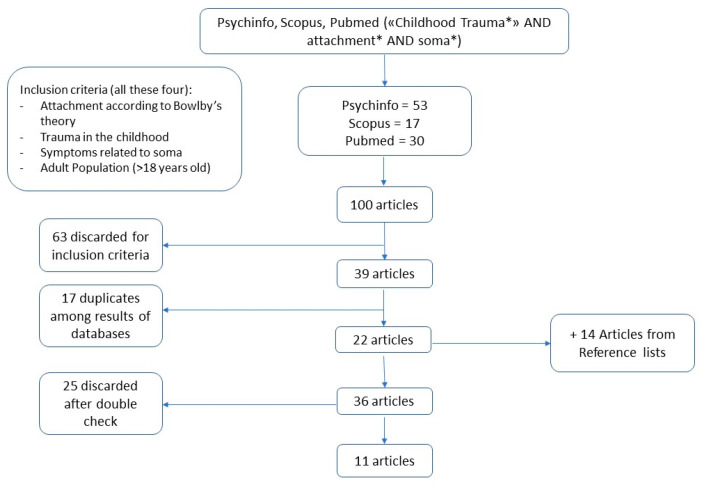
Flow chart of scoping review.

## Data Availability

The data that support the findings of this review are available upon request from the first author [P.S.G.].

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
