# Peer review of "How Does Trauma Make You Sick? The Role of Attachment in Explaining Somatic Symptoms of Survivors of Childhood Trauma"

_healthcare, 2024, doi:10.3390/healthcare12020203_

Round 1
Reviewer 1 Report
Comments and Suggestions for Authors
Dear authors, thank you for your interesting paper that addresses important issues related to childhood experiences and trauma. Despite different aspects of chilhood traumas attract attention of reseachers for quite a long time, still the understanding of these effects needs more data and analysis.
Still, I believe there are some comments that need to be addressed:
1. You never mention in your article the characteristics of the samples, only amount, although it definitely needs to be discussed. The age, gender, professional background etc could be important factors that impacted results and thus to make any conclusions you need to address it. Authors need to describe samples in more details and discuss results in context of samples specifics.
2. It would be great if in discussion or limitation section you put more attention to differences between the scales used. You slightly mention it in limitation section now, but it definitely deserves more attention and discussion. Authors analyze studies that used different methods and I believe this part needs deeper discussion.
These two comments may seem to be not much critiques, but for instance they address basics of this paper. I believe authors can make these clarifications and the paper will sound much more solid.
Author Response
REVIEWER 1
Dear authors, thank you for your interesting paper that addresses important issues related to childhood experiences and trauma. Despite different aspects of chilhood traumas attract attention of reseachers for quite a long time, still the understanding of these effects needs more data and analysis.
Authors: we thank the reviewer for the general positive evaluation and for the strengths highlighted.
Still, I believe there are some comments that need to be addressed:
You never mention in your article the characteristics of the samples, only amount, although it definitely needs to be discussed. The age, gender, professional background etc could be important factors that impacted results and thus to make any conclusions you need to address it. Authors need to describe samples in more details and discuss results in context of samples specifics.
Authors: Samples’ descriptions were included in Table 1, in order to specify sample size, mean age ad gender of the groups analyzed. Some considerations were added in limitation section.
- It would be great if in discussion or limitation section you put more attention to differences between the scales used. You slightly mention it in limitation section now, but it definitely deserves more attention and discussion. Authors analyze studies that used different methods and I believe this part needs deeper discussion.
Authors: The use of different methods to evaluate the constructs was discussed in the limitation section.
These two comments may seem to be not much critiques, but for instance they address basics of this paper. I believe authors can make these clarifications and the paper will sound much more solid.
Authors: we agree with the relevance of the comments and suggestions provided and we hope that the revised version of the manuscript may be improved in scientific soundness.

Reviewer 2 Report
Comments and Suggestions for Authors
Thank you for the opportunity of reviewing the manuscript. The childhood trauma is an issue of great importance. I've got some suggestions regarding the paper itself:
- Have the Authors included other languages, than English? It is not explained directly in the Methodology section. If not, there is a risk, that the review might be incomplete.
- what about The Grey literature?
- Have the Authors used any type of software, like Covidence?
Author Response
REVIEWER 2
Thank you for the opportunity of reviewing the manuscript. The childhood trauma is an issue of great importance. I've got some suggestions regarding the paper itself:
Authors: we thank the reviewer for the general positive evaluation and for the strengths highlighted.
- Have the Authors included other languages, than English? It is not explained directly in the Methodology section. If not, there is a risk, that the review might be incomplete.
Authors: we thank the reviewer for this comment that gives us the possibility to clarify this point. We limited the research to English-language manuscript us common in systematic/narrative review since it may be difficult to totally understand a paper that was not written in a language internationally adopted for international scientific communication. Nevertheless, we added this point in limit section.
- what about The Grey literature?
Authors: we thank the reviewer for this comment, we decided to include only the studies published in the academic area, published on scientific journal and contained within important international database. Nevertheless, some papers or material in the grey literature could have been relevant but the impossibility to have guarantee about the related scientific soundness make us choose to not included this kind of literature. Nevertheless, we added this point in limit section.
- Have the Authors used any type of software, like Covidence?
Authors: we thank the reviewer for this comment that gives us the possibility to clarify this point. Wi do not use any type of software but may be useful for future investigation.

Reviewer 3 Report
Comments and Suggestions for Authors
Greenman and colleagues wrote a scoping review to understand the relationship between ACEs, complex trauma, and physical health outcomes. My main suggestions surround the need to further clarify and define concepts like ACEs, trauma, somatic symptoms, etc., and to consult with a librarian to conduct a thorough, scoping review on this topic. I appreciate the authors' efforts and hope they find this feedback beneficial.
-I would recommend a very thorough look at several of the key terms or concepts the authors used, as this guides their review and research question for the scoping review. Specifically, I would encourage the authors to clarify their use of the term "somatic symptoms." Somatic symptoms are typically thought to be symptoms (with and without readily identifiable physical cause) that cause significant stress and functional impairment to the individual. The authors have provided examples of diseases like obesity, diabetes, cancer, and heart disease as somatic symptoms, which is untraditional. I am curious if by "somatic symptoms" the authors actually meant "physical health outcomes."
-On pg 2, lines 45-46, the authors note that they grouped both ACEs and complex trauma together for their study. The majority of previous research in this area seem to treat these concepts differently. They are related, but not one in the same. Prior research usually treats ACEs as an exposure or event, while trauma or complex trauma is the outcome or consequence of being exposed to the ACEs event. It will be very important to clarify whether they were interested in investigating the link between exposure, attachment, and physical health or trauma, attachment, and physical health.
The introduction would benefit from a more in-depth review of what is currently known about ACEs, trauma, attachment, and physical health, and what is not known that this review will address. I respectfully disagree with the authors that there is a paucity of studies exploring these relationships. A more thorough review of existing research in this area will help to guide their specific research questions. Likewise, more specifically stating the purpose of the study and their research question may clarify any misconceptions about what their work adds to current research.
On pg 2, lines 96-98 the authors made a recommendation for future research. For organization and clarity, it would be helpful to move this to the Future Directions section.
Under Methods, it would be helpful to the reader to explicitly state the research question being addressed by their scoping review. Similarly, it would be very helpful to specifically outline the inclusion and exclusion criteria in the text. Please also discuss the screening process used to include or exclude papers, and the level of agreement amongst reviewers. It would be helpful to also describe the process for extracting data from the paper--was a numerical or thematic analysis conducted? (the scoping review procedure I am familiar with is from Mak & Thomas, 2022, which encourages a scientific look at the reviewed work). I would suggest consulting with a librarian to ensure all the steps of a scoping review process are addressed.
The Results section reviews the findings from each individual paper, but this section does not provide the reader a summation of the results. What are the overarching findings of this review?
Please exercise caution when extrapolating results (for example, on pg 6 beginning line 280, the authors suggested that there is a link between attachment and PNES but the link might be mediated by aspects of attachment not evaluated by the study's authors, such as disorganized attachment--this was not investigated the study they were summarizing).
Similarly, the authors reported that the results of their review suggest a moderating or mediating role in association between childhood trauma and somatic symptoms in adulthood, but their review does not support this conclusion, as they did not conduct this type of analysis (pg 6, line 384)
In the discussion section, the authors stated that the results of their review supported a moderating and mediating effect of attachment on several variables (pg 11, beginning ine 370). However, the authors did not conduct moderating/mediating analyses on the data extracted from their review. This section should be significantly edited to reflect the findings of their review.
The authors introduce Social Baseline Theory and Wilma Bucci's Multiple Code Theory as potential explanations for the relationship between adverse childhood experiences, insecure attachment, and result in risk factors for health concerns. However, these theories were not part of their scoping review, which was conducted in an effort to understand the relationship between ACEs, trauma, insecure attachment, and health concerns. Can the authors clarify if this was part of their review?
When discussing limitations, it is helpful to the reader to be explicit--it is confusing to refer to "well known limitations" (pg 12, line 433) without further explanation.
The authors suggest that future research should use sources other than self-report questionnaires, but then suggested interview (which is another form of self-report). More information on why an interview vs questionnaire format might be more helpful is needed here.
The authors stated: "Now that the possibility of attachment playing a role in the health manifestations of childhood trauma has been established..." (pg 12, beginning line 444). However, this relationship has already been established in previous literature. It would be helpful to really focus on where results of their review uniquely adds to this literature base.
Comments on the Quality of English LanguageSome minor editing for academic tone might be helpful. No other English language editing appears needed.
Author Response
REVIEWER 3
Greenman and colleagues wrote a scoping review to understand the relationship between ACEs, complex trauma, and physical health outcomes. My main suggestions surround the need to further clarify and define concepts like ACEs, trauma, somatic symptoms, etc., and to consult with a librarian to conduct a thorough, scoping review on this topic. I appreciate the authors' efforts and hope they find this feedback beneficial.
Authors: we thank the reviewer for the general positive evaluation and for the strengths highlighted. We agree with the relevance of the comments and suggestions provided and we hope that the revised version of the manuscript may be improved in scientific soundness.
-I would recommend a very thorough look at several of the key terms or concepts the authors used, as this guides their review and research question for the scoping review. Specifically, I would encourage the authors to clarify their use of the term "somatic symptoms." Somatic symptoms are typically thought to be symptoms (with and without readily identifiable physical cause) that cause significant stress and functional impairment to the individual. The authors have provided examples of diseases like obesity, diabetes, cancer, and heart disease as somatic symptoms, which is untraditional. I am curious if by "somatic symptoms" the authors actually meant "physical health outcomes."
Authors: we thank the reviewer for this point that gives us the opportunity to clarify this point. For the purposes of this scoping review, we considered “somatic symptoms” to be any pathological condition, including both physical diseases with a clear physiological aetiology, and “somatic symptom and related disorders” as defined in the fifth edition of the Diagnostic and Statistical Manual of Mental Disorders (DSM-5) (American Psychiatric Association, 2013), which are physical problems that do not have an identifiable physiological origin causing significant stress and functional impairment to the individual. Thus somatic symptoms included all typology of physical health outcomes identified by the respectively studies’ Authors as somatic outcome. The choice of the term “somatic symptoms” was considered to be a good option because it is both suffi-ciently specific and appropriately general to identify studies of a variety of somatic conditions.
-On pg 2, lines 45-46, the authors note that they grouped both ACEs and complex trauma together for their study. The majority of previous research in this area seem to treat these concepts differently. They are related, but not one in the same. Prior research usually treats ACEs as an exposure or event, while trauma or complex trauma is the outcome or consequence of being exposed to the ACEs event. It will be very important to clarify whether they were interested in investigating the link between exposure, attachment, and physical health or trauma, attachment, and physical health.
Authors: we have clarified (p. 2) the focus on exposure to traumatic events, which can lead to distinct psychological syndromes.
The introduction would benefit from a more in-depth review of what is currently known about ACEs, trauma, attachment, and physical health, and what is not known that this review will address. I respectfully disagree with the authors that there is a paucity of studies exploring these relationships. A more thorough review of existing research in this area will help to guide their specific research questions. Likewise, more specifically stating the purpose of the study and their research question may clarify any misconceptions about what their work adds to current research.
Authors: we thank the reviewer for this comment. We have clarified this point explaining that there is a wide literature on these thematic but usually the studies explored the associations between two of the three constructs and a paucity of studies jointly exploring the three dimensions of interest in the present study.
On pg 2, lines 96-98 the authors made a recommendation for future research. For organization and clarity, it would be helpful to move this to the Future Directions section.
Authors: we thank the reviewer for this comment that gives us the possibility to improve the clarity of the paper. We moved the sentence in the correct section, as suggested.
Under Methods, it would be helpful to the reader to explicitly state the research question being addressed by their scoping review. Similarly, it would be very helpful to specifically outline the inclusion and exclusion criteria in the text. Please also discuss the screening process used to include or exclude papers, and the level of agreement amongst reviewers. It would be helpful to also describe the process for extracting data from the paper--was a numerical or thematic analysis conducted? (the scoping review procedure I am familiar with is from Mak & Thomas, 2022, which encourages a scientific look at the reviewed work). I would suggest consulting with a librarian to ensure all the steps of a scoping review process are addressed.
Authors: The research question, inclusion criteria, and screening process are now detailed in the paper (p. 3). Two judges independently evaluated the eligibility of each manuscript and in case of disagreement, a third independent judge evaluated the paper in question until a final agreement was reached. According to your suggestion, we calculated and added the interrater reliability (Cohen’s k = 0.76). We conducted a thematic analysis according to the descriptive nature of the scoping review.
The Results section reviews the findings from each individual paper, but this section does not provide the reader a summation of the results. What are the overarching findings of this review?
Authors: Thank you for this comment. We have added a brief summary of the results at the end of the results section (p. 10).
Please exercise caution when extrapolating results (for example, on pg 6 beginning line 280, the authors suggested that there is a link between attachment and PNES but the link might be mediated by aspects of attachment not evaluated by the study's authors, such as disorganized attachment--this was not investigated the study they were summarizing).
Authors: We thank the reviewer for this comment. The sentence has been reworded to demonstrate more caution.
Similarly, the authors reported that the results of their review suggest a moderating or mediating role in association between childhood trauma and somatic symptoms in adulthood, but their review does not support this conclusion, as they did not conduct this type of analysis (pg 6, line 384)
Authors: We thank the reviewer for this comment. According to reviewer’s suggestion we separate what the studies sustain from the clinical hypothesis that can be proposed for explaining the associations emerging from the narrative review. In this light we can only hypothesize a mediating role according to a clinical- theoretical background.
In the discussion section, the authors stated that the results of their review supported a moderating and mediating effect of attachment on several variables (pg 11, beginning ine 370). However, the authors did not conduct moderating/mediating analyses on the data extracted from their review. This section should be significantly edited to reflect the findings of their review.
Authors: We thank the reviewer for this comment that give us the opportunity to clarify this point. We reworded the first sentence of the paragraph due to its lack of clarity. Indeed, we claim that in the international literature a mediating/moderating role of attachment is plausible, not that this is what can be assumed/sustained by the analysis of the studies that we have conducted. (We can just hypothesize it according to previous literature reivews and theoretical considerations). Therefore, we more clearly report this part to be coherent with the scoping review interpretation in the light of the broader international literature.
The authors introduce Social Baseline Theory and Wilma Bucci's Multiple Code Theory as potential explanations for the relationship between adverse childhood experiences, insecure attachment, and result in risk factors for health concerns. However, these theories were not part of their scoping review, which was conducted in an effort to understand the relationship between ACEs, trauma, insecure attachment, and health concerns. Can the authors clarify if this was part of their review?
Authors: We thank the reviewer for this comment that give us the opportunity to clarify this point. The theories cited were not part of our review’s purpose, however we think that provide some possible theoretical explanation for the associations emerging from the analysis of the studies included in our narrative review may be useful. In this light, to avoid misunderstanding we strongly reduced the part dedicated to these theories, reporting only that MCT may be useful in explaining the link between the investigated constructs.
When discussing limitations, it is helpful to the reader to be explicit--it is confusing to refer to "well known limitations" (pg 12, line 433) without further explanation.
Authors: we thank the reviewer for this comment. We have clarified the limitation associated to self-report evaluation.
The authors suggest that future research should use sources other than self-report questionnaires, but then suggested interview (which is another form of self-report). More information on why an interview vs questionnaire format might be more helpful is needed here.
Authors: we thank the reviewer for this comment. We clarified that we suggest to use clinician-report interviews in which a trained clinician scores the responses produced by participants according to specific validated guidelines, as for the AAI.
The authors stated: "Now that the possibility of attachment playing a role in the health manifestations of childhood trauma has been established..." (pg 12, beginning line 444). However, this relationship has already been established in previous literature. It would be helpful to really focus on where results of their review uniquely adds to this literature base.
Authors: we thank the reviewer for this comment. We reworded the paragraph to be more focused on the scoping review’s findings.
